# NAS-Bench-1Shot1:
# Benchmarking and Dissecting
# One-shot Neural Architecture Search

**Arber Zela**[1*]**, Julien Siems**[1*]**, & Frank Hutter**[1,2]
[1]Department of Computer Science, University of Freiburg
[2]Bosch Center for Artificial Intelligence
{`zelaa, siemsj, fh`}@cs.uni-freiburg.de

## Abstract

One-shot neural architecture search (NAS) has played a crucial role in making NAS methods computationally feasible in practice. Nevertheless, there is still a lack of understanding on how these weight-sharing algorithms exactly work due to the many factors controlling the dynamics of the process. In order to allow a scientific study of these components, we introduce a general framework for one-shot NAS that can be instantiated to many recently-introduced variants and introduce a general benchmarking framework that draws on the recent large-scale tabular benchmark NAS-Bench-101 for cheap anytime evaluations of one-shot NAS methods. To showcase the framework, we compare several state-of-the-art one-shot NAS methods, examine how sensitive they are to their hyperparameters and how they can be improved by tuning their hyperparameters, and compare their performance to that of blackbox optimizers for NAS-Bench-101.

## 1 Introduction

While neural architecture search (NAS) has attracted a lot of attention due to the effectiveness in automatically designing state-of-the-art neural networks (Zoph & Le, 2017; Zoph et al., 2018; Real et al., 2017; 2019), the focus has recently shifted to making the search process more efficient (Pham et al., 2018; Elsken et al., 2019; Liu et al., 2019; Xie et al., 2019; Cai et al., 2019; Casale et al., 2019). The most crucial concept which led to a reduction in search costs to the order of a single function evaluation is certainly the weight-sharing paradigm: Training only a single large architecture (the *one-shot* model) subsuming all the possible architectures in the search space (Brock et al., 2018; Pham et al., 2018).

Despite the great advancements of these methods, the exact results of many NAS papers are often hard to reproduce (Li & Talwalkar, 2019; Yu et al., 2020; Yang et al., 2020). This is a result of several factors, such as unavailable original implementations, differences in the employed search spaces, training or evaluation pipelines, hyperparameter settings, and even pseudorandom number seeds (Lindauer & Hutter, 2019). One solution to guard against these problems would be a common library of NAS methods that provides primitives to construct different algorithm variants, similar to what as RLlib (Liang et al., 2017) offers for the field of reinforcement learning. Our paper makes a first step into this direction.

Furthermore, experiments in NAS can be computationally extremely costly, making it virtually impossible to perform proper scientific evaluations with many repeated runs to draw statistically robust conclusions. To address this issue, Ying et al. (2019) introduced NAS-Bench-101, a large tabular benchmark with 423k unique cell architectures, trained and fully evaluated using a one-time extreme amount of compute power (several months on thousands of TPUs), which now allows to cheaply simulate an arbitrary number of runs of NAS methods, even on a laptop. NAS-Bench-101 enabled a comprehensive benchmarking of many discrete NAS optimizers (Zoph & Le, 2017; Real et al., 2019), using the exact same settings. However, the discrete nature of this benchmark does not

---

*Equal contribution

allow to directly benchmark one-shot NAS optimizers (Pham et al., 2018; Liu et al., 2019; Xie et al., 2019; Cai et al., 2019). In this paper, we introduce the first method for making this possible.

Specifically, after providing some background (Section 2), we make the following contributions:

1. We introduce *NAS-Bench-1Shot1*, a novel benchmarking framework that allows us to reuse the extreme amount of compute time that went into generating NAS-Bench-101 (Ying et al., 2019) to cheaply benchmark one-shot NAS methods. Our mapping between search space representations is novel to the best of our knowledge and it allows querying the performance of found architectures from one-shot NAS methods, contrary to what is claimed by Ying et al. (2019). Specifically, it allows us to follow the full trajectory of architectures found by arbitrary one-shot NAS methods at each search epoch without the need for retraining them individually, allowing for a careful and statistically sound analysis (Section 3).

2. We introduce a general framework for one-shot NAS methods that can be instantiated to many recent one-shot NAS variants, enabling fair head-to-head evaluations based on a single code base (Section 4).

3. We use the above to compare several state-of-the-art one-shot NAS methods, assess the correlation between their one-shot model performance and final test performance, examine how sensitive they are to their hyperparameters, and compare their performance to that of black-box optimizers used in NAS-Bench-101 (Section 5).

We provide our open-source implementation[1], which we expect will also facilitate the reproducibility and benchmarking of other one-shot NAS methods in the future.

## 2 BACKGROUND AND RELATED WORK

### 2.1 NAS-BENCH-101

NAS-Bench-101 (Ying et al., 2019) is a database of an exhaustive evaluation of all architectures in a constrained cell-structured space on CIFAR-10 (Krizhevsky, 2009). Each cell is represented as a directed acyclic graph (DAG) where the nodes represent operation choices and the edges represent the information flow through the neural network (see also Figure 1 and Section 3.1). To limit the number of architectures in the search space, the authors used the following constraints on the cell: 3 operations in the operation set $\mathcal{O} = \{$3x3 convolution, 1x1 convolution, 3x3 max-pool$\}$, at most 7 nodes (this includes input and output node, therefore 5 choice nodes) and at most 9 edges.

These constraints, and exploiting symmetries, reduced the search space to 423k unique valid architectures. Each architecture was trained from scratch three times to also obtain a measure of variance. In addition, each architecture was trained for 4, 12, 36 and 108 epochs; for our analysis, we mainly used the results for models trained for 108 epochs, if not stated otherwise.

### 2.2 NAS-BENCH-102

Concurrently to this work, Dong & Yang (2020) released NAS-Bench-102, which is another NAS benchmark that, differently from NAS-Bench-101, enables the evaluation of weight-sharing NAS methods. Their search space consists of a total of 15,625 architectures, which is exhaustively evaluated on 3 image classification datasets. Similarly to Zela et al. (2020) and this work, Dong & Yang (2020) found that architectural overfitting occurs for DARTS for all their datasets.

While NAS-Bench-102 and this work go towards the same direction, they differ in many ways:

1. They use extensive computation to create a *new* benchmark (with 15.625 architectures), while we devise a novel reformulation to reuse the even much more extensive computation of the NAS-Bench-101 dataset ( 120 TPU years) to create three new one-shot search spaces with the larges one containing 363.648 architectures. This required zero additional computational cost.

2. We show that it *is* possible to reuse the graph representation in NAS-Bench-101 to run one-shot NAS methods; this requires changes to the one-shot search space, but allows a mapping which can be used for architecture evaluation.

---

[1]https://github.com/automl/nasbench-1shot1

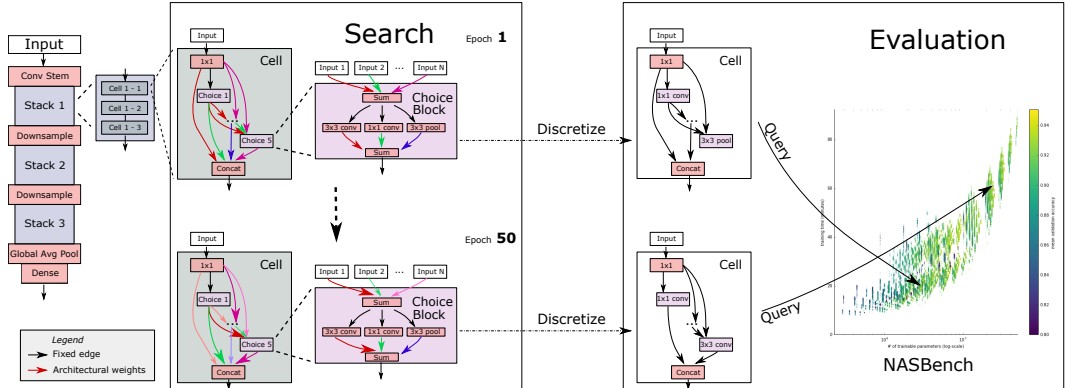

Figure 1: Overview of the NAS-Bench-1Shot1 analysis strategy. The one-shot model we construct only contains discrete architectures that are elements of NAS-Bench-101 (Ying et al., 2019). The cell architecture chosen is similar to that of Bender et al. (2018), with each choice block containing an operation decision. Note that NAS-Bench-101 does not contain a separate reduction cell type. Plot on the right from Ying et al. (2019) (Best viewed in color).

3. They evaluate their search space on 3 image classification datasets, while we introduce 3 different search spaces (as sub-spaces of NAS-Bench-101) with growing complexity.

## 2.3 ONE-SHOT NEURAL ARCHITECTURE SEARCH

The NAS problem can be defined as searching for the optimal operation (e.g. in terms of validation error of architectures) out of the operation set $\mathcal{O}$ in each node of the DAG and for the best connectivity pattern between these nodes.

Designing architectures for optimized accuracy or to comply with resource constraints led to significant breakthroughs on many standard benchmarks (Pham et al., 2018; Zoph & Le, 2017; Brock et al., 2018; Liu et al., 2019; Cai et al., 2019; Elsken et al., 2019). While early methods were computationally extremely expensive (Zoph & Le, 2017), the weight-sharing paradigm (Brock et al., 2018; Pham et al., 2018) led to a significant increase in search efficiency. Here, the weights of the operations in each architecture are shared in a supermodel (the so-called *one-shot model* or convolutional neural fabric (Saxena & Verbeek, 2016)), which contains an exponential number of sub-networks, each of which represents a discrete architecture. Architectures whose sub-networks share components (nodes/edges) also share the weights for these components' operations; therefore, in analogy to DropOut (Srivastava et al., 2014), training one architecture implicitly also trains (parts of) an exponential number of related architectures. There are a variety of methods on how to conduct NAS by means of the one-shot model (Brock et al., 2018; Pham et al., 2018; Bender et al., 2018; Liu et al., 2019; Li & Talwalkar, 2019) (see also Appendix B), but the final problem is to find the optimal sub-network in this one-shot model.

The weight sharing method was used to great effect in DARTS (Liu et al., 2019), where it allows a gradient based optimization of both the architectural and the one-shot weights. Subsequent work on DARTS has addressed further lowering the computational and the memory requirements (Dong & Yang, 2019; Xu et al., 2020; Cai et al., 2019; Casale et al., 2019).

One fundamental drawback of the weight sharing method is the fact that the architecture search typically takes place in a lower fidelity model (e.g., using less cells and/or cheaper operations): the so-called *proxy model*. After the search, a discrete architecture is derived from the proxy model which is then trained with more parameters — a stage often referred to as *architecture evaluation*. This poses the question whether the architecture found in the proxy model is also a good architecture in the bigger model, a question studied by several recent works (Bender et al., 2018; Yu et al., 2020).

## 3 A GENERAL FRAMEWORK FOR BENCHMARKING ONE-SHOT NAS

We will now introduce our framework for cheaply benchmarking the anytime performance of one-shot NAS methods. Our main analysis strategy is the following: First, we run the search procedure of various methods and save the architecture weights of the one-shot models for each epoch. Second, we find the discrete architecture at each epoch and query it in NAS-Bench-101. The last step is not trivial due to the different representations of the search space used in NAS-Bench-101 and standard one-shot methods. Ying et al. (2019) state that one-shot methods cannot be directly evaluated on NAS-Bench-101. In the following sections we present a mapping between these different search space representations, which eventually enable us to evaluate one-shot methods on NAS-Bench-101. To the best of our knowledge this is a novel contribution of this paper.

### 3.1 SEARCH SPACE REPRESENTATION

In order to carry out the analysis we propose in this work, we had to construct a search space that only contains discrete architectures that are also contained in NAS-Bench-101. This allows us to look up any discrete architectures' performance in NAS-Bench-101 when the larger model is trained from scratch. Unfortunately, this is non-trivial since the NAS-Bench-101 space does not match the typical space used in one-shot NAS methods. We separately consider the various parts of the search space.

**Network-Level Topology.** In terms of network-level topology, our search spaces closely resemble the models which were evaluated in NAS-Bench-101. We used the same macro architecture as in NAS-Bench-101, i.e., 3 stacked blocks with a max-pooling operation in-between, where each block consists of 3 stacked cells (see Figure 1). While our final evaluation models exactly follow NAS-Bench-101 in order to be able to reuse its evaluations, our one-shot model only has 16 initial convolution filters, rather than the 128 used in NAS-Bench-101. This is a common practice to accelerate NAS and used similarly in, e.g., Liu et al. (2019).

**Cell-Level Topology.** The cell-level structure is represented as a DAG, where the input node is the output of a previous cell or the convolutional stem, and the output node is a concatenation of all the previous nodes. In order to have the operation choices still in the intermediate nodes of the DAG, we adapt the *choice block* motif from Bender et al. (2018) as depicted in Figure 1. The edges connecting input, output nodes and choice blocks represent only the information flow in the graph. To have a large and expressive enough search space(s), we introduce the following architectural weights in the DAG edges:

- $\alpha^{i,j}$ to edges connecting nodes $i < j$ to choice block $j$. The input of choice block $j$ is then computed as $I^j = \sum_{i<j} \frac{\exp(\alpha^{i,j})}{\sum_{i'<j} \exp(\alpha^{i',j})} x^i$, where $x^i$ is the output tensor of node $i$ (either input node or choice block).

- $\gamma^{j,k}$ to the edges connecting the input node or choice blocks $j < k$ to the output node $k$ of the cell, where the corresponding feature maps are concatenated: $O^k = \oplus_{j<k} \frac{\exp(\gamma^{j,k})}{\sum_{j'<k} \exp(\gamma^{j',k})} x^j$, where $\oplus$ is the concatenation operator.

Note that the non-linearity applied to the edge weights varies depending on the NAS optimizer used; e.g. for GDAS (Dong & Yang, 2019) and SNAS (Xie et al., 2019) it would be a Gumbel-Softmax (Eric Jang & Poole, 2017) instead.

**Choice Blocks.** As in Bender et al. (2018), each choice block inside the cell can select between the operations in the operations set $\mathcal{O}$ of NAS-Bench-101. In order to find the optimal operation in each choice block via gradient-based one-shot NAS methods, we assign an architectural weight $\beta^o$ to each operation $o \in \mathcal{O}$ inside the choice block. The output of the choice block $j$ is computed by adding element-wise the latent representations coming from the operations outputs:

$$x^j = \sum_{o \in \mathcal{O}} \frac{\exp(\beta^o)}{\sum_{o' \in \mathcal{O}} \exp(\beta^{o'})} o(I^j), \tag{1}$$

which is basically the so-called *MixedOp* in DARTS. NASBench cells contain 1x1 projections in front every operation (demonstrated in Figure 1 in (Ying et al., 2019)). The number of output channels of each projection is chosen such that the output has the same number of channels as the

input. This adaptive choice for the number of channels is incompatible with the one-shot model due to the different tensor dimensionality coming from previous choice blocks. We used 1x1 projections with a fixed number of channels instead.

## 3.2 EVALUATION PROCEDURE

By means of these additional weights we do not restrict the possible architectures in the search space to contain only a fixed number of edges per cell, as done for example in Zoph et al. (2018), Pham et al. (2018), Liu et al. (2019), etc. This requirement would have restricted our architectural decisions heavily, leading to only small search spaces.

Table 1 shows the characteristics of each search space. We propose three different search spaces by making different decisions on the number of parents each choice block has. The decisions affect the quality and quantity of the architectures contained in each search space. For all search spaces note that the sum of the number of parents of all nodes in the search space is chosen to be 9, to match the NAS-Bench-101 requirement. Search space 1, 2 and 3 have 6240, 29160 and 363648 architectures with loose ends respectively, making search space 3 the largest investigated search space. To the best of our knowledge search space 3 is currently the largest and only available tabular benchmark for one-shot NAS. For details on each search space see Appendix A.

Table 1: Characteristic information of the search spaces.

|  |  | Search space | | |
| --- | --- | --- | --- | --- |
|  |  | 1 | 2 | 3 |
| No. parents | Node 1 | 1 | 1 | 1 |
|  | Node 2 | 2 | 1 | 1 |
|  | Node 3 | 2 | 2 | 1 |
|  | Node 4 | 2 | 2 | 2 |
|  | Node 5 | - | - | 2 |
|  | Output | 2 | 3 | 2 |
| No. archs. | w/ loose ends | 6240 | 29160 | 363648 |
|  | w/o loose ends | 2487 | 3609 | 24066 |

Given the architectural weights of the cell shown in Figure 1 we query the test and validation error of the discrete architecture from NAS-Bench-101 as follows.

1. We determine the operation chosen in each choice block by choosing the operation with the highest architectural weight.

2. We determine the parents of each choice block and the output by choosing the top-$k$ edges according to Table 1 (e.g. for choice block 4 in search space 3 we would choose the top-2 edges as parents).

3. From 1. we construct the operation list and from 2. the adjacency matrix of the cell which we use to query NAS-Bench-101 for the test and validation error.

Each node in the graph chooses its parents during evaluation following e.g. DARTS (Liu et al., 2019). However, because edges model information flow and the output edges are also architectural decisions there is possibility of a node being a loose end. These are nodes whose output does not contribute to the output of the discrete cell, as seen in the upper cell under evaluation of Figure 1. As a result, we can count the number of architectures with or without loose ends. Note, that had we chosen the children of each node we could have invalid architectures where a node has an output but no input.

## 4 A GENERAL FRAMEWORK FOR ONE-SHOT NAS METHODS

Most of the follow-up works of DARTS (Algorithm 1), which focus on making the search even more efficient and effective, started from the original DARTS codebase[2], and each of them only change very few components compared to DARTS.

---

[2]https://github.com/quark0/darts

**Algorithm 1** DARTS

1: $I^j = \sum_{i<j} S(\alpha^{i,j}) x^i$
2: $O^k = \oplus_{j<k} S(\gamma^{j,k}) x^j$
3: $x^j = \sum_{o \in \mathcal{O}} S(\beta^o) o(I^j)$
4: m ← DAG($I^j, O^k, x^j$)
5: **while** not converged **do**
6:      m.update($\Lambda, \nabla_\Lambda \mathcal{L}_{valid}$)
7:      m.update($w, \nabla_w \mathcal{L}_{train}$)
8: **end while**
9: **Return** $\Lambda$

**Algorithm 2** PC-DARTS

1: $I^j = \sum_{i<j} S(\alpha^{i,j}) x^i$
2: $O^k = \oplus_{j<k} S(\gamma^{j,k}) x^j$
3: $x^j = \sum_{o \in \mathcal{O}} S(\beta^o) o(\boxed{M^o * I^j}) \boxed{+(1 - M^o * I^j)}$
4: m ← DAG($I^j, O^k, x^j$)
5: **while** not converged **do**
6:      m.update($\Lambda, \nabla_\Lambda \mathcal{L}_{valid}$)
7:      m.update($w, \nabla_w \mathcal{L}_{train}$)
8: **end while**
9: **Return** $\Lambda$

**Algorithm 3** GDAS

1: $I^j = \sum_{i<j} GS(\alpha^{i,j}) x^i$
2: $O^k = \oplus_{j<k} GS(\gamma^{j,k}) x^j$
3: $x^j = \sum_{o \in \mathcal{O}} GS(\beta^o) o(I^j)$
4: m ← DAG($I^j, O^k, x^j$)
5: **while** not converged **do**
6:      m.update($\Lambda, \nabla_\Lambda \mathcal{L}_{valid}$)
7:      m.update_single_path($w, \nabla_w \mathcal{L}_{train}$)
8: **end while**
9: **Return** $\Lambda$

**Algorithm 4** Random NAS with Weight-sharing

1: $I^j = \sum_{i<j} x^i$
2: $O^k = \oplus_{j<k} x^j$
3: $x^j = \sum_{o \in \mathcal{O}} o(I^j)$
4: m ← DAG($I^j, O^k, x^j$)
5: **while** not converged **do**
6:      arch ← sample_uniformly_at_random(m)
7:      m.update_weights_of_single_architecture(arch, $w, \nabla_w \mathcal{L}_{train}$)
8:
9: **end while**
10: **for** i ∈ 1..1000 **do**
11:      arch_samples ← sample_uniformly_at_random(m)
12: **end for**
13: **Return** arch ∈ arch_samples with lowest validation error

**Algorithm 5** ENAS

1: $I^j = \sum_{i<j} x^i$
2: $O^k = \oplus_{j<k} x^j$
3: $x^j = \sum_{o \in \mathcal{O}} o(I^j)$
4: m ← DAG($I^j, O^k, x^j$)
5: **while** not converged **do**
6:      arch ← sample_using_controller(m)
7:      m.update_weights_of_single_architecture(arch, $w, \nabla_w \mathcal{L}_{train}$)
8:      Update RNN controller
9: **end while**
10: **for** i ∈ 1..100 **do**
11:      arch_samples ← sample_using_controller(m)
12: **end for**
13: **Return** arch ∈ arch_samples with lowest validation error

Algorithm 2 and Algorithm 3 highlight these components (relative to DARTS) for PC-DARTS (Xu et al., 2020) and GDAS (Dong & Yang, 2019), respectively. For example, when comparing PC-DARTS and DARTS, the only difference in our benchmark is the partial channel connections (line 3 of Algorithm 2) in the choice blocks, which consists of a channel sampling mask $M^o$ that drops feature maps coming from $I^j$. GDAS, on the other hand, replaces the Softmax (*S*) function in DARTS by a Gumbel-Softmax (*GS*), which applies for every architectural weight in $\Lambda = \{\alpha, \beta, \gamma\}$ (lines 1-3 in Algorithm 3), and uses this concrete distribution to sample single paths through the cell during search (line 7 in Algorithm 3). Random Search with Weight Sharing (Random WS) (Li & Talwalkar, 2019) (Algorithm 4) and ENAS (Pham et al., 2018) (Algorithm 5) do not need the continuous relaxation in order to conduct the architecture search, instead they sample randomly in RandomWS or from the recurrent neural network controller (line 6 in Algorithm 1) in ENAS, in order to select the sub-network in the one-shot model to train.

These close correspondences between current one-shot NAS variants provide an opportunity to implement all of these variants in the same general code basis. This allows us to (a) automatically guard against any confounding factors when evaluating the strengths and weaknesses of different approaches, and (b) allows us to mix and match the components of different algorithms. We implemented all variants in a single code basis, which we are committed to grow into a flexible library of primitives for one-shot NAS methods, and for which we will gladly accept any help the community wants to provide.

One-shot NAS methods in this code basis inherit all the methods and attributes necessary for building the one-shot computational graph from a base parent class. This encapsulation and modularity ensures that all differences in their performance come from a few lines of code, and that all other confounding factors cannot affect these results. This will also facilitate the incorporation of other one-shot NAS methods and pinpoint the components that differ in them.

Furthermore, the primitives encoding the search spaces presented in Section 3 are defined separately from the NAS optimizers. This encapsulation will allow researchers to study each of these components in isolation, experimenting with one of them while being sure that the other one does not change.

## 5 NAS-BENCH-1SHOT1 AS A BENCHMARK AND ANALYSIS FRAMEWORK

We now demonstrate the use of NAS-Bench-1Shot1 as a benchmark for one-shot NAS. We first evaluate the anytime performance of five different one-shot NAS methods: DARTS (Liu et al., 2019), GDAS (Dong & Yang, 2019), PC-DARTS (Xu et al., 2020), ENAS (Pham et al., 2018) and Random Search with Weight Sharing (Random WS) (Li & Talwalkar, 2019).[3] Afterwards, we investigate the robustness of these one-shot NAS optimizers towards their search hyperparameters and show that if these hyperparameters are carefully tuned, the one-shot NAS optimizer can outperform a wide range of other discrete NAS optimizers in our search spaces.

### 5.1 COMPARISON OF DIFFERENT ONE-SHOT NAS OPTIMIZERS

We ran the NAS search for 50 epochs[4] using their respective default hyperparameter settings (see Appendix C). If not stated otherwise, all the following results were generated by running each experiment with six random seeds (0 to 5). All plots show the mean and standard deviation of the test regret queried from NAS-Bench-101. Over the three independent trainings contained in NASBench-101 for each architecture on each epoch budget we average. The search was done on a single NVIDIA RTX2080Ti using the same python environment. In Figure 2 we report the anytime test regret for each of these methods. Our findings can be summarized as follows:

- While the one-shot validation error converged in all cases (except for GDAS due to the temperature annealing) the queried test error from NAS-Bench-101 of the architectures increased at some point, indicating that the architectural parameters overfit the validation set. This phenomenon occurred frequently for DARTS. The same result was also previously observed by Zela et al. (2020) on subspaces of the standard DARTS space.

- PC-DARTS demonstrated both a stable search and relatively good final performance in all search spaces, notably the best overall performance for search space 3. We attribute this behaviour to the regularization effect present in PC-DARTS via the partial channel connections (see Zela et al. (2020) and Section 5.3).

- Random WS and ENAS mainly explore poor architectures across all three search spaces. This behaviour is a result of the small correlation between the architectures evaluated with the one-shot weights and their true performance during architecture evaluation (as queried from NAS-Bench-101 (see Section 5.2)). This correlation directly affects these methods since in the end of search they sample a certain number of architectures (1000 randomly sampled for Random WS and 100 using the learned controller policy for ENAS) and evaluate them using the one-shot model weights in order to select the architecture which is going to be trained from scratch in the final evaluation phase. When running ENAS for 100 epochs in search space 2 (Figure 7 in the appendix) we see that the it performs better than Random WS. ENAS also has a stronger correlation between the sampled architectures and the NAS-Bench-101 architectures for search space 2 (see Section 5.2).

- GDAS performs quite robustly across all 3 benchmarks, however, due to the temperature annealing of the Gumbel Softmax, it might manifest some premature convergence to a sub-optimal local minimum.

### 5.2 CORRELATION ANALYSIS

Many one-shot NAS methods, such as ENAS (Pham et al., 2018), NAONet (Luo et al., 2018) or Random WS (Li & Talwalkar, 2019) select the final architecture by means of the one-shot parameters. In order to assess if this is optimal we computed the correlation between a given architecture's one-shot test error and its respective NAS-Bench-101 test error, for all 4 available budgets in NAS-Bench-101 on every 10th search epoch. This analysis was performed for all architectures without loose ends in each search space. The only exception is ENAS, for which we decided to evaluate the correlation by sampling 100 architectures (as done by the algorithm after the search has finished to select the one to retrain from scratch) from the controller instead of evaluating every architecture in the search space.

---

[3]Details on these optimizers can be found in Appendix B.
[4]This required different amounts of time: DARTS 1st order: 3h, DARTS 2nd order: 4h, GDAS: 1.5h, PC-DARTS: 2h, ENAS: 4h, Random-WS: 11h.

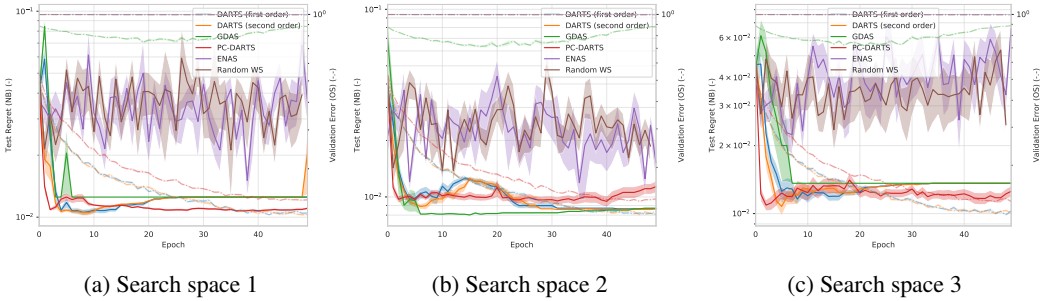

(a) Search space 1      (b) Search space 2      (c) Search space 3

Figure 2: Comparison of different one-shot NAS optimizers on the three different search spaces defined on NASBench. The solid lines show the anytime test regret (mean ± std), while the dashed blurred lines the one-shot validation error (Best viewed in color).

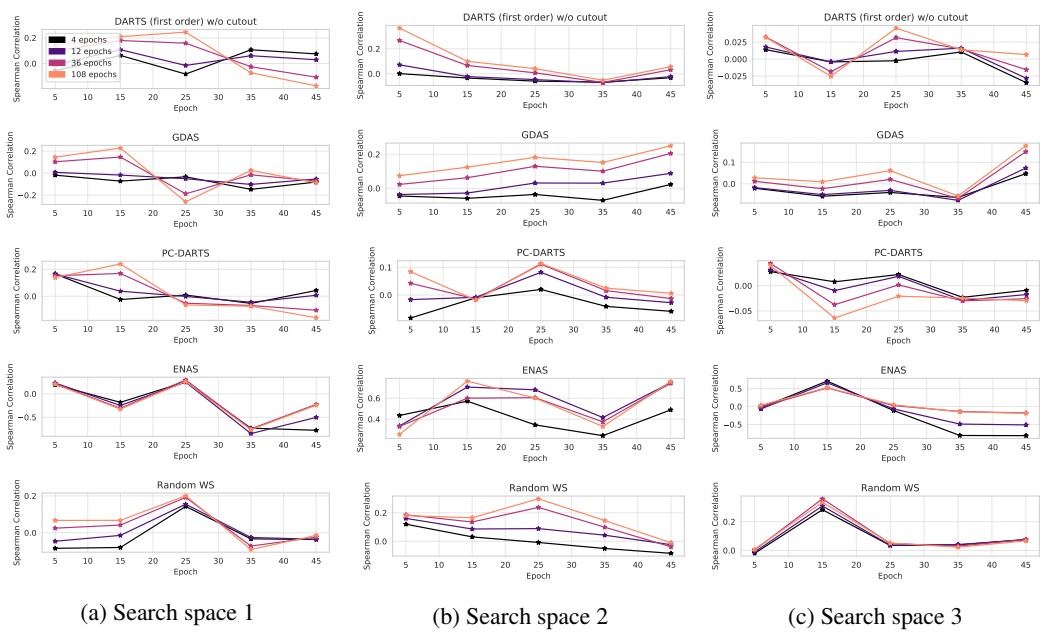

(a) Search space 1      (b) Search space 2      (c) Search space 3

Figure 3: Correlation between the one-shot validation error and the corresponding NAS-Bench-101 test error for each search space. (Best viewed in color).

As shown in Figure 3, there is almost no correlation between the weight sharing ranking and the true one (Spearman correlation coeff. between -0.25 and 0.3) during search for DARTS, PC-DARTS, GDAS and Random WS. Only ENAS shows some correlation for search space 2 and some anticorrelation for search spaces 1 and 3. These results agree with the ones reported by Yu et al. (2020) (who could only do this evaluationx on a small search space) and explain the poor performance of Random WS and ENAS on our benchmarks, since the architectures sampled during evaluation and ranked according to their one-shot validation error are unlikely to perform well when evaluated independently. To the best of our knowledge this is the first time that an evaluation of this correlation is conducted utilizing such a large number of architectures in the search space, namely 24066 different architectures for search space 3. We added further experiments on the correlation between the lower fidelity proxy model used in architecture search and the final model from architecture evaluation in the Appendix H.

## 5.3 ROBUSTNESS OF ONE-SHOT NAS OPTIMIZERS

As already observed by Zela et al. (2020), DARTS tends to become more robust when the inner objective $\mathcal{L}_{train}$ in the bi-level optimization procedure has a relatively strong regularization factor

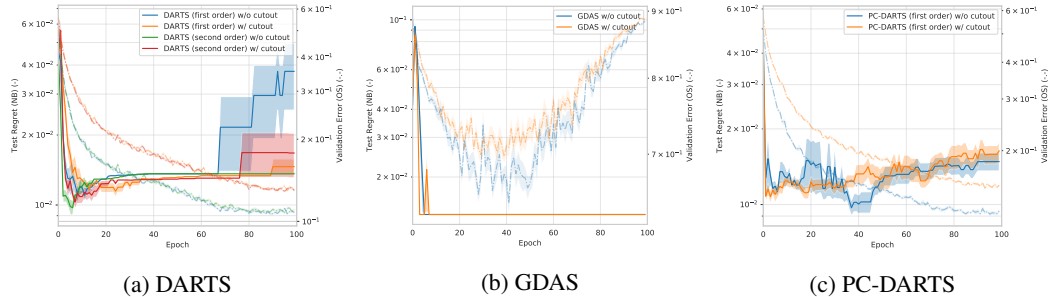

(a) DARTS         (b) GDAS         (c) PC-DARTS

Figure 4: Illustration of the impact that Cutout has on the test regret on NAS-Bench-101 and the validation error of the one-shot model using DARTS, GDAS and PC-DARTS on search space 3 (Best viewed in color).

during search. In order to investigate the long term behaviour, for this analysis we chose to run every search for 100 epochs instead of 50 epochs. Similarly to Zela et al. (2020), we find that enabling Cutout (DeVries & Taylor, 2017) or increasing the $L_2$ factor for the search model weights, has a substantial effect on the quality of the solutions found by the NAS optimizers. See Figure 4 for the Cutout (CO) results and Figure 11 (in the appendix) for the results with different $L_2$ regularization. Based on these results we observe that:

- As DARTS overfits to the validation set at the end of search for search space 1 and 3, applying Cutout during search either keeps the solutions to a good local minimum or reduces the overfitting effect, a finding similar as the one in Zela et al. (2020).

- Interestingly, while Zela et al. (2020) only showed overfitting behavior for DARTS, it may also occur for GDAS (Figure 13 in the appendix) and PC-DARTS (Figure 14 in the appendix), which indicates that this might be an intrinsic property of these methods due to the local updates in the architecture space.

- In PC-DARTS, there is already a strong regularization effect as a result of the partial channel connectivity, which explains the robust behavior of this optimizer and that its results deteriorate on average as we increase $L_2$ regularization.

### 5.4 TUNABILITY OF ONE-SHOT NAS HYPERPARAMETERS

Next to the hyperparameters used in the evaluation pipeline, one-shot NAS methods have several hyperparameters of their own, such as the regularization hyperparameters studied in Section 5.3, learning rates, and other hyperparameters of the search phase.

Naively tuning these hyperparameters with the one-shot validation error as the objective would lead to sub-optimal configurations, since, as we saw, this metric is not a good indicator of generalization. In our proposed benchmarks, we can tune these hyperparameters of the NAS optimizer to minimize the validation error queried from NAS-Bench-101. By doing so, we aim to shed more light onto the influence these hyperparameters have during the one-shot search, and to study the sensitivity of the NAS method towards these hyperparameters.

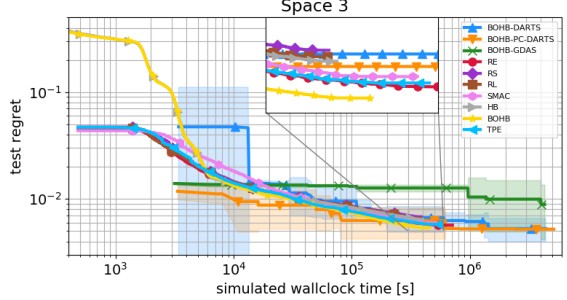

Figure 5: Optimizing the hyperparameters of one-shot optimizers with BOHB on search space 3. (best viewed in color). Results for search space 1 and 2 are shown in Figure 16.

To this end, we constructed 3 configuration spaces of increasing cardinality, CS1, CS2 and CS3 (see Appendix F for details), which only include hyperparameters controlling the NAS process. We chose BOHB (Falkner et al. (2018), see Appendix F for details) as the hyperparameter

optimization method and DARTS as our NAS method to be tuned across all configuration spaces, since in our benchmarks it was more brittle than PC-DARTS and GDAS. We provide the results in Appendix F.2.

In order to compare the tunability of NAS methods, we used BOHB to optimize all of DARTS, PC-DARTS, and GDAS, starting from their respective default hyperparameter settings. Figure 5 shows the anytime test regret of the architectures found by the respective NAS method's configurations tried by BOHB; as the figure shows, PC-DARTS and GDAS start with much more robust hyperparameter settings, but DARTS can also be tuned to perform as well or better. We note that carrying out this optimization on NAS-Bench-1Shot1 reduced the time for tuning DARTS from a simulated 45 GPU days to 1 day on 16 GPUs.

Figure 5 also provides an evaluation of DARTS, GDAS and PC-DARTS compared to the state-of-art discrete NAS optimizers used by Ying et al. (2019) (such as RL, RE, and HPO methods). Since these one-shot NAS methods are much faster than black-box methods, it is possible to tune them *online* using BOHB and the resulting BOHB-{DARTS, GDAS, PC-DARTS} typically still yields better performance over time. Note that we never use the validation set split used to evaluate the individual architectures in NAS-Bench-101 during the architecture search. This subset with 10k examples is only used to compute the objective function value that BOHB optimizes. Therefore, the one-shot optimizers use 20k examples for training and 20k for the architectural parameter updates. The x-axis in Figures 5 shows the *simulated wall-clock time*: $t_{sim} = t_{search} + t_{train}$, where $t_{search}$ is the time spent during search by each configuration and $t_{train}$ is the training time for 108 epochs (queried from NAS-Bench-101) of the architectures selected by the one-shot optimizer. From this experiment, we make the following observations:

- On all search spaces the architectures BOHB-DARTS found outperformed the architectures found by the default DARTS configuration by a factor of 7 to 10.

- The found robust configurations did not only avoid overfitting in the architectural level, but they also typically outperformed the architectures found by state-of-art discrete NAS optimizers used by Ying et al. (2019) (such as RL, RE, and HPO methods).

- The multi-fidelity in BOHB does not only accelerate the hyperparameter optimization procedure, but in this case also allows to determine the sufficient number of epochs to run the NAS optimizer in order to get an optimal architecture. In fact, the best configuration on each of the incumbents comes usually from the lowest budget, i.e. 25 search epochs.

- The most robust configuration on each of the search spaces are typically the ones with a large regularization factor, relative to the default value in Liu et al. (2019) (see Figure 19 in the appendix).

## 6  CONCLUSION AND FUTURE DIRECTIONS

We proposed NAS-Bench-1Shot1, a set of 3 new benchmarks for one-shot neural architecture search which allows to track the trajectory and performance of the found architectures computationally cheaply. Using our analysis framework, we compared state-of-the-art one-shot NAS methods and inspected the robustness of the methods and how they are affected by different hyperparameters. Our framework allows a fair comparison of any one-shot NAS optimizer and discrete NAS optimizers without any confounding factors. We hope that our proposed framework and benchmarks will facilitate the evaluation of existing and new one-shot NAS methods, improve reproducibility of work in the field, and lead to new insights on the underlying mechanisms of one-shot NAS.

## ACKNOWLEDGMENTS

The authors acknowledge funding by the Robert Bosch GmbH, support by the European Research Council (ERC) under the European Unions Horizon 2020 research and innovation program through grant no. 716721, and by BMBF grant DeToL.

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

## A   DETAILS ON SEARCH SPACES

**Search space 1**   The main characteristic of this search space is that the number of parents for each choice block and output has to be exactly 2 (apart from choice block 1 which is only connected to the input). Because of this requirement one choice block had to be discarded as that would exceed the requirement to have at most 9 edges. The total distribution of test error in shown in Figure 6a. It is the smallest search space discussed in this work.

**Search space 2**   This search space is related to search space 1 in that it only consists of 4 intermediate nodes, but in contrast the output has three parents and nodes 1 and 2 only one parent. This increases the number of architectures in this space compared to search space 1. The test error distribution is shown in Figure 6b.

**Search space 3**   All 5 intermediate nodes are used in this search space, making this search space the largest, but also the search space where each node has on average the least number of parents. The test error distribution is shown in Figure 6c.

## B   OPTIMIZERS

**DARTS (Liu et al., 2019)** uses a weighted continuous relaxation over the operations to learn an architecture by solving a bilevel optimization problem. The training dataset is split in two parts, one used for updating the parameters of the operations in the one-shot model, and the other to update the weights appended to operations, that determine the importance of that operation.

For evaluation, we choose the parents of each choice block based on the highest architectural weights and the number of parents for that choice block given by Table 1. We pick the highest weighted operation from the choice block.

**GDAS (Dong & Yang, 2019)** modifies DARTS, such that individual paths are sampled differentiably through each cell using Gumbel-Softmax (Eric Jang & Poole, 2017) to adapt the architecture weights. This reduces the memory overhead created by DARTS as only the sampled paths have to be evaluated. GDAS uses the same search space and evaluation procedure as DARTS.

**PC-DARTS (Xu et al., 2020)** reduces the memory overhead by only evaluating a random fraction of the channels with the mixed-ops. The authors argue that this also regularizes the search as it lowers the bias towards weight-free operations such as skip-connect and max-pooling, which are often preferred early on in DARTS search. In addition to partial channel connections, the authors propose edge normalization, which adds additional architectural parameters to the edges connecting to an intermediate node. This is done to compensate for the added fluctuations due to the partial channel connections. These additional weights are already part of the search spaces we proposed in this paper.

**Random Search with Weight Sharing (Random WS) (Li & Talwalkar, 2019)** randomly samples architectures from the one-shot model for each training mini-batch and trains only the selected subset of the one-shot model on that mini-batch. Differently from DARTS, PC-DARTS and GDAS,

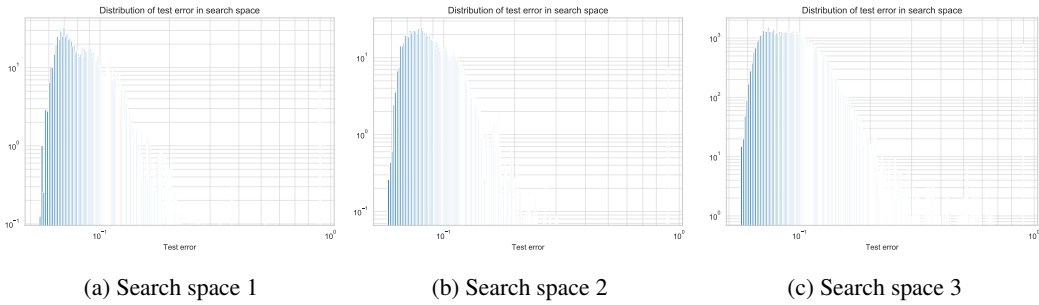

|   (a) Search space 1   |   (b) Search space 2   |   (c) Search space 3   |

Figure 6: Distribution of test error in the search spaces with loose ends.

Random WS does not require a validation set, since there are no architectural weights that need to be updated. For evaluation Random WS samples 1000 architectures from the search space and evaluates each for only a small number of batches on the validation set using the optimized weights of the one-shot model corresponding to the sub-networks. Then the 5 architectures with the lowest one-shot validation error are chosen and fully evaluated on the validation set. The overall best architecture is returned.

**ENAS (Pham et al., 2018)** similarly to Random WS samples sub-networks of in the one-shot model, however by means of a recurrent neural network (RNN) controller rather than randomly. As the search progresses the parameters of the RNN controller are updated via REINFORCE (Williams et al., 2000) using the validation error of the sampled architectures as a reward. This way the sampling procedure is handled in a more effective way.

## C    HYPERPARAMETERS

If not stated otherwise the following hyperparameters were used for all our experiments. We used a batch size of 96 throughout for DARTS, GDAS and PC-DARTS as the search spaces are small enough to allow it and as this reduces the randomness in the training, which makes the comparison between optimizers easier. Random WS was trained with a batch size of 64. All other hyperparameters were adapted from DARTS (Liu et al., 2019).

## D    COMPARISON OF OPTIMIZERS OVER DIFFERENT BUDGETS

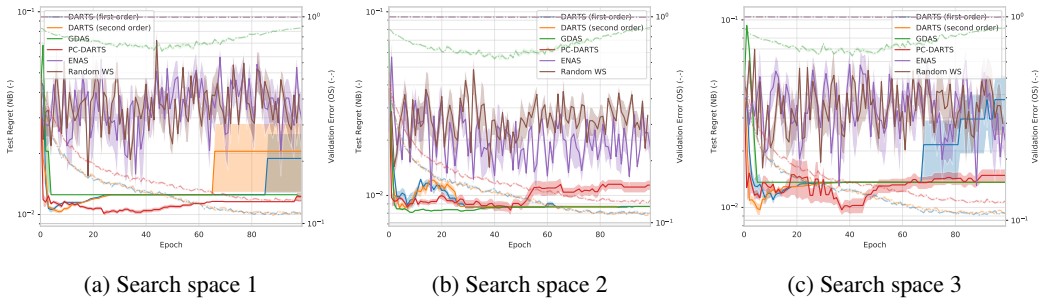

(a) Search space 1          (b) Search space 2          (c) Search space 3

Figure 7: Comparison of different One-Shot Neural Architecture optimizers on the three different search spaces defined on NAS-Bench-101 over 100 epochs.

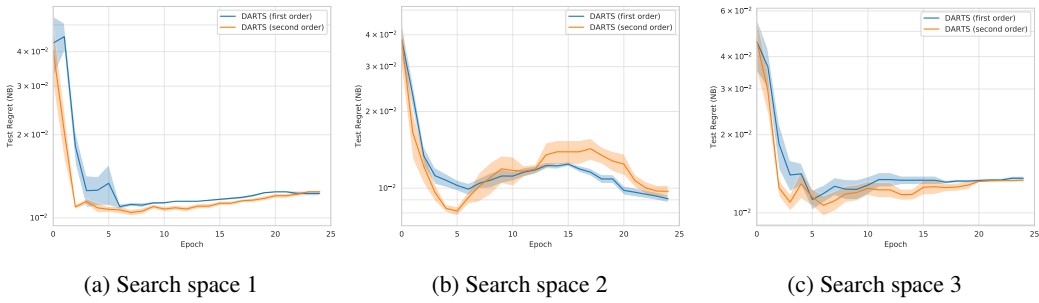

(a) Search space 1          (b) Search space 2          (c) Search space 3

Figure 8: Comparison of DARTS first and second order on the three different search spaces defined on NAS-Bench-101 for 25 epochs.

# E  REGULARIZATION

## E.1  CUTOUT

Interestingly, for GDAS the validation error of the one-shot model is closer linked to the test regret on NASBench as that is the case for DARTS as shown in Figure 9. This is particularly striking in search space 2 in which the local minimum attained by the one-shot validation error is well aligned with the minimum of the test regret. It is interesting to note that GDAS very quickly overfits on this search space, since the validation error increases usually at around 50 epochs. This may be related to the linearly decreasing temperature schedule for the gumbel-softmax (Eric Jang & Poole, 2017) from 10 to 1 as proposed by Dong & Yang (2019). As the temperature decreases the operations with higher architectural weights will be sampled more often leading to overfitting on the architectural level as demonstrated by the increasing one-shot validation error. Cutout has little impact on the search phase of GDAS (Figure 9).

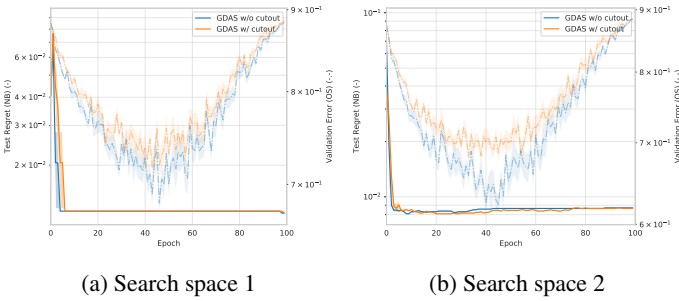

(a) Search space 1 (b) Search space 2

Figure 9: Comparison of the effect of using Cutout during architecture search on GDAS for search space 1 and 2.

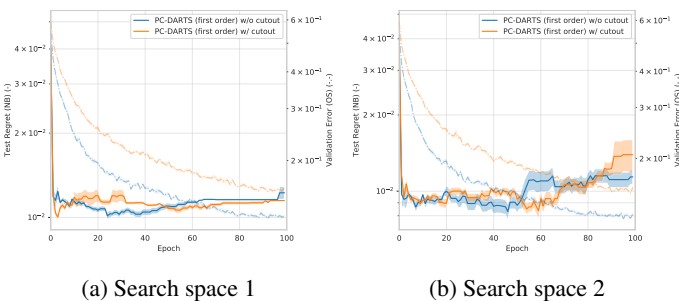

(a) Search space 1 (b) Search space 2

Figure 10: Comparison of the effect of using cutout during architecture search on PC-DARTS for search space 1 and 2.

For PC-DARTS (Figure 10) cutout regularization helped find better architectures in particular in search space 3 and 1. This underlines the fact that strong regularization on the architectural level via partial channel connections can be effectively supported by cutout regularization on the training loss. Second order optimization as proposed by DARTS has no significant benefit for PC-DARTS and often decreases the performance.

## E.2  $L_2$ REGULARIZATION

Increasing the $L_2$ regularization has a positive effect on the found architectures for GDAS in search space 1 and 2 as shown in Figure 13. However, in search space 3 setting the weight decay to $81e^{-4}$ has the effect of making the model unstable.

For PC-DARTS lowering the $L_2$ regularization had overall a positive effect across all search spaces (Figure 14). However, this made the training also less stable as demonstrated by search space 1 (Figure 14a)

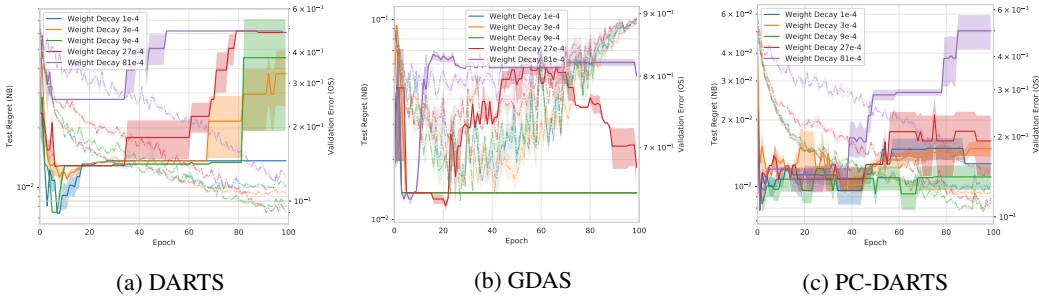

(a) DARTS            (b) GDAS            (c) PC-DARTS

Figure 11: Illustration of the impact that weight decay has on the test regret on NAS-Bench-101 and the validation error of the one-shot model using DARTS, GDAS and PC-DARTS on search space 3 (Best viewed in color).

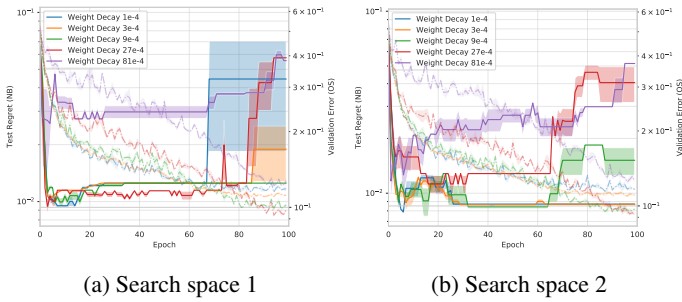

(a) Search space 1            (b) Search space 2

Figure 12: DARTS first order w/o cutout trained with different levels of $L_2$ regularization for search space 1 and 2.

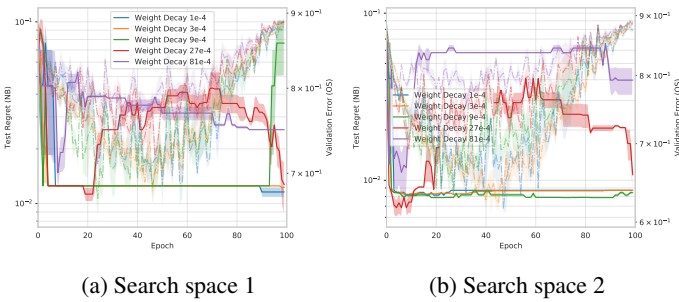

(a) Search space 1            (b) Search space 2

Figure 13: Comparison of the effect of using different values of weight decay during architecture search on GDAS for search space 1 and 2.

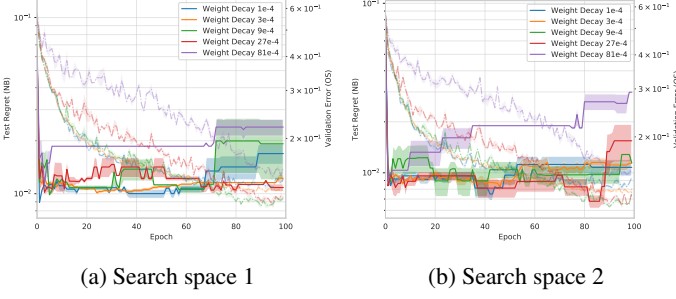

(a) Search space 1            (b) Search space 2

Figure 14: Comparison of the effect of using different values of weight decay during architecture search on PC-DARTS for search space 1 and 2.

## F  BOHB DETAILS

BOHB (Falkner et al., 2018) is a combination of Bayesian Optimization (BO) and Hyperband (HB) (Li et al., 2017). Hyperband uses SuccesssiveHalving (SH) (Jamieson & Talwalkar, 2016) to stop poorly performing trainings early. Success Halving starts trainings with an initial budget and advances the top fraction $(1/\eta)$ of them to the next stage with $\eta$ higher budget. Hyperband uses this a subroutine to evaluate many uniformly at random sampled configurations on small budgets. The budgets and scaling factors are chosen such that all SuccessiveHalving evaluations take approximately the same time. BOHB combines Hyperband with Bayesian Optimization by using a probabilistic model to guide the search towards better configurations. As a result, BOHB performs as well as Hyperband during early optimization, but samples better configurations once enough samples are available to build a model.

### F.1  SETUP

We ran BOHB for 64 iterations of *SuccessiveHalving* (Jamieson & Talwalkar, 2016) on 16 parallel workers, resulting in 280 full function evaluations. In our experiments we use the number of epochs that the one-shot NAS optimizers run the search as the fidelity used by BOHB and optimize the validation error after 108 epochs of training queried from NAS-Bench-101. Namely, we use $min\_budget = 25$ epochs, $max\_budget = 100$ epochs and $\eta = 2$ in BOHB. Note that this is only the number of epochs used for the architecture search. Additionally, we never use the validation set split used to evaluate the individual architectures in NAS-Bench-101 during the architecture search. Therefore, each one-shot NAS optimizer will use 20k examples for training and 20k for search. The x-axis in Figures 15, 16, 17, 18 shows the *simulated wall-clock time*: $t_{sim} = t_{search} + t_{train}$, where $t_{search}$ is the time spent during search by each NAS algorithm configuration and $t_{train}$ is the training time for 108 epochs (queried from NAS-Bench-101) of the architectures selected by the NAS optimizers.

We build 3 configuration spaces with different cardinality and which include hyperparameters affecting the architecture search process. The spaces are as follows:

1. $CS1 = \{L_2, CO\_prob\}$
2. $CS2 = \{L_2, CO\_prob, lr\}$
3. $CS3 = \{L_2, CO\_prob, lr, moment, CO\_len, batch\_size, grad\_clip, arch\_lr, arch\_L_2\}$

### F.2  RESULTS

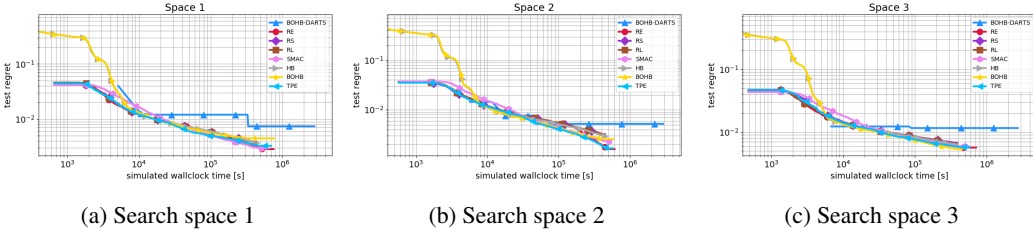

(a) Search space 1          (b) Search space 2          (c) Search space 3

Figure 15: Test regret of architectures found with DARTS (1st order) configurations sampled by BOHB on CS1. All the lines except the BOHB-DARTS one show the mean±std of the best architecture from 500 search repetitions (Best viewed in color).

Interestingly, optimizing on CS2 led to not only a more robust configuration of DARTS, but also in outperforming a state-of-the-art discrete NAS optimizer such as Regularized Evolution (RE) (Real et al., 2019). The found solutions by BOHB also outperform every other one-shot optimizer used throughout this paper with their default settings. Including the learning rate in the configuration space was crucial to achieve such a performance. Figure 16 shows the results when running BOHB with the same settings on CS1. Note that none of the sampled configurations outperforms RE. On the other hand, increasing the cardinality of the configuration space requires many more samples to build a good model. Figure 17 shows the results when optimizing with BOHB on CS3. Even though

the learning rate was inside this space, again none of the sampled configurations is better than the discrete NAS optimizers.

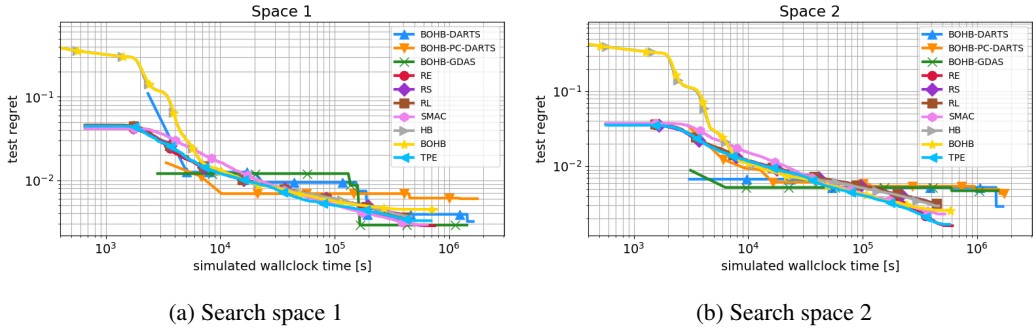

Figure 16: Test regret of architectures found with DARTS, GDAS and PC-DARTS (1st order) configurations sampled by BOHB on CS2. All the lines except BOHB-DARTS, BOHB-GDAS and BOHB-PC-DARTS show the mean±std of the best architecture from 500 search repetitions (Best viewed in color).

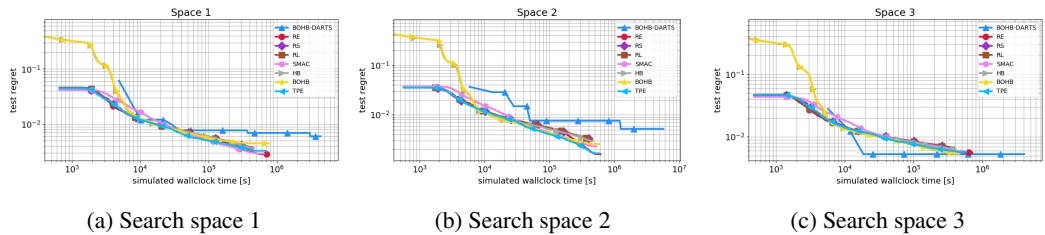

Figure 17: Analogous to Figure 15, with the only difference being that here we optimize on CS3.

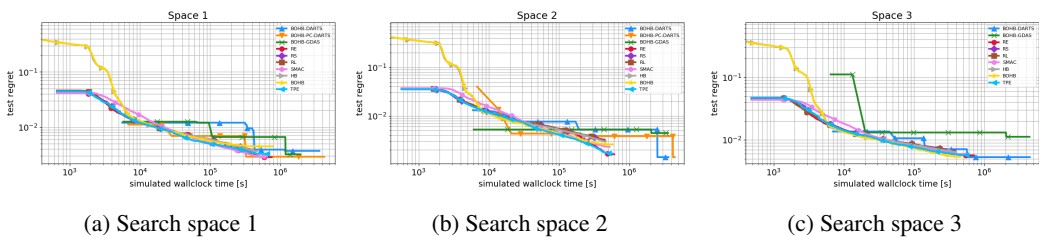

Figure 18: Test regret of architectures found with DARTS, GDAS and PC-DARTS (2nd order) configurations sampled by BOHB on CS2.

### F.3 TRANSFERABILITY BETWEEN SPACES.

The following Table 2 shows the performance of the best found configuration on search space 3 for 50 epochs by BOHB when transferred to search spaces 1 and 2. The results show the mean and standard deviation of the architectures found by 6 independent search runs with the respective optimizers. We can see that there is no clear pattern on what is transferable where.

## G    HYPERPARAMETER IMPORTANCE

To better understand the configuration space which we evaluated with BOHB we use functional analysis of variance (fANOVA) (Hutter et al., 2014). The idea is to assess the importance of individual hyperparameters by marginalizing performances over all possible values other hyperparameters could have taken. The marginalization estimates are determined by a random forest model which

Table 2: Results of architectures found on search space 1 and 2 with the best found configuration for 50 epochs by BOHB on search space 3.

| Optimizer | | Test regret | |
|---|---|---|---|
| | | Search space 1 | Search Space 2 |
| DARTS | Default config. | 1.252e-2 $\pm$ 0.0 | **0.864e-2** $\pm$ 0.023e-2 |
| | Transferred config. | **1.045e-2** $\pm$ 0.171e-2 | 0.992e-2 $\pm$ 0.211e-2 |
| GDAS | Default config. | **1.252e-2** $\pm$ 0.0 | 0.871e-2 $\pm$ 0.0 |
| | Transferred config. | 3.093e-2 $\pm$ 3.975e-2 | **0.831e-2** $\pm$ 0.045e-2 |
| PC-DARTS | Default config. | **1.104e-2** $\pm$ 0.088e-2 | 1.133e-2 $\pm$ 0.404e-2 |
| | Transferred config. | 5.843e-2 $\pm$ 4.118e-2 | **0.992e-2** $\pm$ 0.087e-2 |

was trained on all configurations belonging to specific budgets during the BOHB optimization procedure.

Figure 19 shows the interaction between the Cutout (CO) and $L_2$ factor when optimizing on CS2, DARTS 1st order, for search space 1, 2 and 3. It should be noted that the best found configuration involves at least one relatively high value of one of the regularizers in CS2.

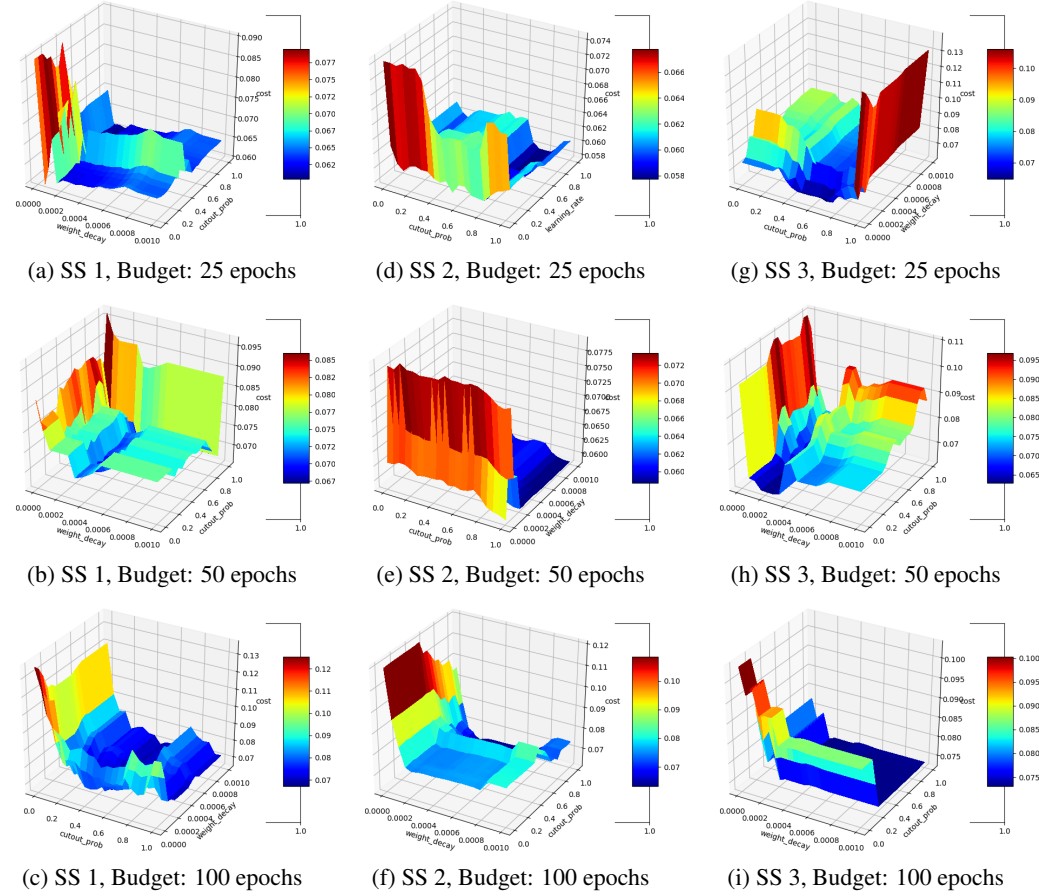

(a) SS 1, Budget: 25 epochs    (d) SS 2, Budget: 25 epochs    (g) SS 3, Budget: 25 epochs

(b) SS 1, Budget: 50 epochs    (e) SS 2, Budget: 50 epochs    (h) SS 3, Budget: 50 epochs

(c) SS 1, Budget: 100 epochs    (f) SS 2, Budget: 100 epochs    (i) SS 3, Budget: 100 epochs

Figure 19: Parameter importance for two hyperparameters, Cutout (CO) and $L_2$ regularization (CS2) across different training epochs and search spaces (SS).

# H CORRELATION BETWEEN THE ARCHITECTURE SEARCH MODEL AND THE ARCHITECTURE EVALUATION MODEL

As a further experiment we wanted to test how strongly the validation error of the models used in architectures search and architecture evaluation are correlated. For this we sampled 150 architectures from search space 3 and trained this architecture in the proxy model. During training we compute the Spearman rank correlation between the validation error of the proxy model and the full architecture evaluation as queried from NAS-Bench-101. The results are shown in Figure 20. Note that 9 cells in the proxy was used for all of our previous experiments as it is also used by the NAS-Bench-101 models.

For 9 cells (Figure 20c) in the proxy model, we find that increasing the total number of channels leads to stronger correlation between the proxy model and the full architecture. However, increasing it beyond 16 channels leads to a decrease in correlation. For 3 cells (Figure 20a) the strongest anytime correlation was interestingly found using only 2 initial channels, with more channels leading to worse performance at the beginning and no better final performance. The results suggest that there exists a set of good combinations between the number of cells and the initial number of channels to get the maximum correlation between the search and evaluation model. As a future work we plan to investigate this relationship, which could eventually lead to a more effective bandit-based NAS method.

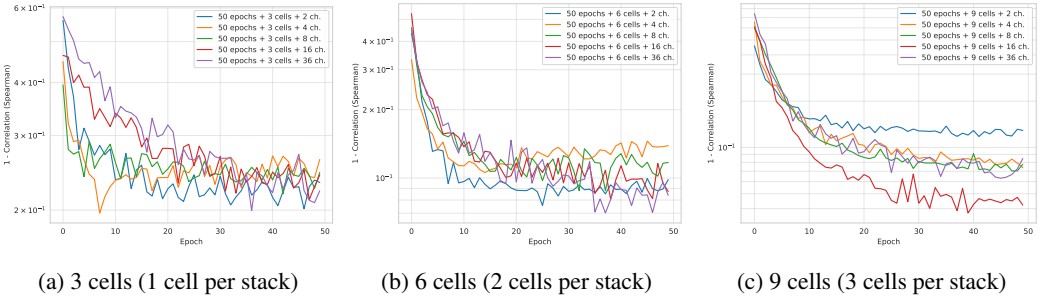

(a) 3 cells (1 cell per stack)  (b) 6 cells (2 cells per stack)  (c) 9 cells (3 cells per stack)

Figure 20: In this experiment we varied the total number of cells within $[3, 6, 9]$ and the number of initial channels of the proxy model within $[2, 4, 8, 16, 36]$.

