# OpenReview forum: "NAS-Bench-1Shot1: Benchmarking and Dissecting One-shot Neural Architecture Search"
_ICLR.cc/2020/Conference — Accept (Poster)_

### Official Review · AnonReviewer1 · 2019-10-10
**Official Blind Review #1**

**Rating:** 1

**Review:**

In this submission, the authors present a benchmark NAS-Bench-1Shot1 for one-shot Network Architecture Search. The presented benchmark reuses the existing NAS-Bench-101 that only contains discrete architectures, and the authors give a method to discretize architectures to match NAS-Bench-101. Also, the authors claim that they introduce a general framework for one-shot NAS methods.

Overall speaking, the technique contribution and novelty of this submission requires further improvement, and I suggest to rejecting this submission. The reasons are detailed as follows:

1) The main contribution of this submission is the benchmark NAS-Bench-1Shot1, which can benefit future research for one-shot NAS. However, this benchmark is based on NAS-Bench-101, and the difference is that NAS-Bench-101 only contains discrete architecture, while the presented NAS-Bench-1Shot1 has a component to discretize architectures and then match to NAS-Bench-101. The novelty may not be enough.

2) The authors claim that they introduce a general framework for one-shot NAS methods. Personally, I think this is over-claimed. There are several variants of DARTS, and the authors just implement these variants and DARTS itself in a unified code base. It would be a true general framework if it can also work with other one-shot NAS methods such as ENAS and the rest ones.

Given these, this submission requires further improvement, especially in terms of technique contribution and novelty.

One interesting thing is that the authors try to use HPO method such as BOHB to tune the hyperparameter of NAS methods.

**Experience Assessment:**

I have read many papers in this area.

**Review Assessment: Checking Correctness Of Derivations And Theory:**

N/A

**Review Assessment: Checking Correctness Of Experiments:**

I carefully checked the experiments.

**Review Assessment: Thoroughness In Paper Reading:**

I read the paper thoroughly.

---

> ### Author Response · Authors · 2019-11-13
> **Response to Reviewer #1 (1/2)**
>
> We would like to thank the reviewer for his/her feedback. We address the concerns raised by the reviewer in the following paragraphs:
>
> > The main contribution of this submission is the benchmark NAS-Bench-1Shot1, which can benefit future research for one-shot NAS. However, this benchmark is based on NAS-Bench-101 [1], and the difference is that NAS-Bench-101 only contains discrete architecture, while the presented NAS-Bench-1Shot1 has a component to discretize architectures and then match to NAS-Bench-101. The novelty may not be enough.
>
> We agree with the reviewer that NAS-Bench-101 is not new, but we argue that our paper is nevertheless novel, because:
> - Finding a way to reuse NAS-Bench-101 is one of the strengths of our approach, since the immense computational resources required to generate NAS-Bench-101 (120 TPU years!) can be efficiently reused this way. We believe this is indeed *more* novel than to simply create another benchmark, and it is far more environmentally friendly since it comes with zero additional computational burden. As we demonstrate in our paper our benchmark allows for a quick and thorough analysis of different NAS methods, which would otherwise be too time consuming.
>
> - The mapping between different search space representations is novel to the best of our knowledge. Most previous work in NAS represents the architecture search space as a directed acyclic graph, however they differ by where the operation choices are encoded in the graph -- either nodes (e.g. ENAS [2]) or edges (e.g. DARTS [3]). By introducing a framework that allows us to create search spaces that automatically satisfy all the constraints in NAS-Bench-101, one can clearly investigate the contribution that each specific algorithm has on the achieved performance, without adding a confounding factor due to a different search space.
>
> - A correlation analysis between the stand-alone architectures evaluated with the one-shot weights and retrained from scratch has not been done before for all the architectures in such a large search space (more than 360k architectures).
>
> - As the reviewer already pointed out, the tunability of one-shot NAS optimizers has not been addressed before.
>
> > The authors claim that they introduce a general framework for one-shot NAS methods. Personally, I think this is over-claimed. There are several variants of DARTS, and the authors just implement these variants and DARTS itself in a unified code base. It would be a true general framework if it can also work with other one-shot NAS methods such as ENAS and the rest ones.
>
> We acknowledge that at the time of submission we had focused on implementing various NAS methods based on DARTS, due to the interest which it has created in the community. However, while the code base of Random NAS with weight sharing [4] is based on DARTS, its search method differs and already demonstrates that similar sampling-based search methods can also fit into our framework. (Indeed, our framework allows to give further clues on why the results of RandomWS were subpar, by demonstrating the weak correlation between one-shot model and final architecture performance.) We have now integrated ENAS (Algorithm 5 in the paper) in our framework and updated the figures in the paper accordingly. Thanks for this suggestion! We are currently working on integrating other methods, such as ProxylessNAS [5] or SNAS [6].
>
> Note that our claim that NAS-Bench-1Shot1 is a general framework for one-shot NAS methods (and not only; also for NAS methods using black-box optimizers) is also due to our novel unification between different graph representations in existing NAS papers.
>
> One other advantage of using the NAS-Bench-101 benchmark is the evaluation of all the architectures using different budgets. This allows evaluating the sensitivity of NAS algorithms during search with respect to the number of epochs used to train individual architectures for estimating the validation error returned to the NAS optimizer (e.g., one can expect a different behavior of ENAS if the sampled architectures from the controller would be evaluated using the lowest or the highest training budget in NAS-Bench-101). This also allows running multi-fidelity black-box optimizers, such as Hyperband [7] or BOHB [8], in our framework.
>
> > One interesting thing is that the authors try to use HPO method such as BOHB to tune the hyperparameter of NAS methods.
>
> Thanks! We also updated the results with more HPO runs.
>
> We hope that with this response we addressed all your concerns and that you will consider updating your assessment, particularly seeing that we have now demonstrated that our framework also encompasses ENAS. Thank you for your time and effort.

---

> > ### Author Response · Authors · 2019-11-13
> > **Response to Reviewer #1 (2/2)**
> >
> > -- References --
> > [1] Chris Ying, Aaron Klein, Esteban Real, Eric Christiansen, Kevin Murphy, Frank Hutter, NAS-Bench-101: Towards Reproducible Neural Architecture Search, ICML 2019
> > [2] Hieu Pham, Melody Y. Guan, Barret Zoph, Quoc V. Le, Jeff Dean, Efficient Neural Architecture Search via Parameter Sharing, ICML 2018
> > [3] Hanxiao Liu, Karen Simonyan, Yiming Yang, DARTS: Differentiable Architecture Search, ICLR 2019
> > [4] LIAM LI, AMEET TALWALKAR. Random Search and Reproducibility for Neural Architecture Search, ArXiv 2019
> > [5] Han Cai, Ligeng Zhu, Song Han, ProxylessNAS: Direct Neural Architecture Search on Target Task and Hardware, ICLR 2019
> > [6] Sirui Xie, Hehui Zheng, Chunxiao Liu, Liang Lin, SNAS: Stochastic Neural Architecture Search, ICLR 2019
> > [7] Lisha Li, Kevin Jamieson, Giulia DeSalvo, Afshin Rostamizadeh, Ameet Talwalkar, Hyperband: A Novel Bandit-Based Approach to Hyperparameter Optimization, JMLR 2018
> > [8] Stefan Falkner, Aaron Klein, Frank Hutter, BOHB: Robust and Efficient Hyperparameter Optimization at Scale, ICML 2018

---

### Official Review · AnonReviewer2 · 2019-10-25
**Official Blind Review #2**

**Rating:** 8

**Review:**


This paper proposes a benchmark dataset for evaluating One-Short Neural Architecture Search models. It extends on the idea of the NASBench-101 (Ying et. al., 2019) into One-Shot models.

I think this is important work. With all the development of neural archictecture search methods, reproducible research is paramount. This is even more timely considering the fact that these problems are computationally expensive, so it is even more computationally difficult than usual to run exhaustive comparisons among proposed methods.

The paper is well-written and the design decisions are clearly explained. The comparison of NAS methods is also interesting to read. In general, I think this is a worthy publication.


**Experience Assessment:**

I have read many papers in this area.

**Review Assessment: Checking Correctness Of Derivations And Theory:**

I assessed the sensibility of the derivations and theory.

**Review Assessment: Checking Correctness Of Experiments:**

I assessed the sensibility of the experiments.

**Review Assessment: Thoroughness In Paper Reading:**

I read the paper at least twice and used my best judgement in assessing the paper.

---

> ### Author Response · Authors · 2019-11-13
> **Response to Reviewer #2**
>
> We greatly appreciate the reviewer's effort and the very positive review. We share the reviewer’s hope that NAS-Bench-1Shot1 will serve the community to prototype and benchmark NAS algorithms without confounding factors.

---

### Official Review · AnonReviewer4 · 2019-11-01
**Official Blind Review #4**

**Rating:** 8

**Review:**

This paper introduces a benchmarking framework to evaluate a class of one-shot NAS methods by exploiting the NAS-Bench-101 database. In order to do so, the authors apply techniques from Bender et al. (2018) to match the search spaces in NAS-Bench to one-shot scenarios and weight edges in the search graph allowing for a variable number of edges per cell. These weights are then used to query a larger, discrete architecture from NAS-Bench for the corresponding evaluation errors. The authors motivate the soundness of a unique framework by showing minimal differences between three NAS optimizers.
In their analysis, they also corroborate the findings of Sciuto et al. (2019) on a larger scale, showing little generalization of the ranking from one-shot validation to the final architecture. Finally, the authors also use this framework to show that a fine-tuned baseline compares favorably to SOTA methods, and that large regularization factors lead to more robust configurations.

The paper is well-written and introduces a framework to cheaply compare one-shot NAS optimizers based on expensive computations behind the NAS-Bench-101 benchmark. In addition, the authors provide empirical analyses that reflect generalization and regularization issues with current methods and that could lead towards designing more robust algorithms.

In my opinion, the the background section might be improved by presenting higher level notions that would lead to an easier understanding by a wider audience. Also, although comprehensible due to the nature of this study, readers not familiar with a variety of previous work might find difficult to parse the paper as many concepts are just referenced.

**Experience Assessment:**

I do not know much about this area.

**Review Assessment: Checking Correctness Of Derivations And Theory:**

I assessed the sensibility of the derivations and theory.

**Review Assessment: Checking Correctness Of Experiments:**

I assessed the sensibility of the experiments.

**Review Assessment: Thoroughness In Paper Reading:**

I read the paper thoroughly.

---

> ### Author Response · Authors · 2019-11-13
> **Response to Reviewer #4**
>
> We thank the reviewer for reading our paper thoroughly and for the very positive feedback.
>
> > the background section might be improved
> Based on the reviewer’s suggestion we updated Section 2 in the paper and added some more details for each respective NAS method we used throughout the paper in Appendix B. We hope that now the concepts will be easier to understand, also for a reader not familiar with the different NAS paradigms, and without going through the literature in detail.

---

### Author Response · Authors · 2019-11-13
**General comment to the AC and to reviewers**

We thank the reviewers for reading our paper thoroughly and for their feedback. We updated the paper as follows:
- More independent runs for each method in all the experiments in the paper.
- Added ENAS [1] (Algorithm 5 in the paper) to the codebase and benchmarked it in all our search spaces. The results for this optimizer are shown in Figure 2 and Figure 7 in the paper.
- Added more detailed information to the background section.
- Fixed a small bug in the codebase and updated the plots with the new results. The paper’s analysis and conclusions of course remain unchanged. In order to make the results reproducible, we provide all the seeds and settings with the code release.

We would like to note that there was a parallel concurrent work submitted to ICLR 2020 named: “An Algorithm-Agnostic NAS Benchmark”. We believe this is also a very valuable benchmark, and it also received very positive scores (8,8,6); the reason we point this out here is that we believe the reviewers’ arguments for giving high scores also apply to our paper. In particular, they emphasize the importance of good benchmarks like these for one-shot NAS algorithms in the reproducibility crisis. The main differences between the two benchmarks are as follows:

- They use extensive computation to create a *new* benchmark (with 15.625 architectures), while we devise a novel reformulation to reuse the even much more extensive computation of the NAS-Bench-101 dataset (~120 TPU years) to create three new one-shot search spaces with 6240, 29.160, and 363.648 architectures each.
- We believe that our reuse of the extreme computation of NAS-Bench-101 is environmentally friendly, since it provides similar value as a new benchmark at zero computational cost.
- We show that (contrary to their claim), it *is* possible to reuse the graph representation in NAS-Bench-101 to run one-shot NAS methods; this requires changes to the one-shot search space, but allows a mapping which can be used for architecture evaluation.
- They evaluate their search space on 3 image classification datasets, while we introduce 3 different search spaces (as sub-spaces of NAS-Bench-101) with growing complexity.
- Our paper does not only serve as a benchmark, but also provides additional insights into the NAS optimization process, such as:
      1. Investigating the correlation between the stand-alone models evaluated with the one-shot weights vs. retrained from scratch (to the best of our knowledge, the first time this is done for a large search space (more than 360k architectures).
      2. Analyzing the anytime performance of one-shot NAS algorithms
      3. Tuning the hyperparameters these optimizers such that they perform comparable with black-box NAS optimizers with the same number of function evaluations.

We thank the reviewers and the AC for their time and effort for ICLR.

-- References --
[1] Hieu Pham, Melody Y. Guan, Barret Zoph, Quoc V. Le, Jeff Dean, Efficient Neural Architecture Search via Parameter Sharing, ICML 2018

---

### Public Comment · ~Gladis_Ne_Limes1 · 2023-12-26
**re**

Protecting sensitive data is a top priority. Establishing stringent security protocols, including secure communication channels and data encryption, is crucial to mitigate data security risks.
https://mlsdev.com/services/it-staff-augmentation

---

### Decision · Program_Chairs · 2019-12-19

**Decision:**

Accept (Poster)

**Comment:**

The authors present a new benchmark for architecture search. Reviews were somewhat mixed, but also with mixed confidence scores. I recommend acceptance as poster - and encourage the authors to also cite https://openreview.net/forum?id=HJxyZkBKDr